

# Mechanical work extraction from an error-prone active dynamic Szilard engine

**Luca Cocconi[1⋆], Paolo Malgaretti[2†] and Holger Stark[3‡]**

**1** Max Planck Institute for Dynamics and Self-Organization, Göttingen 37073, Germany
**2** Helmholtz Institut Erlangen-Nürnberg for Renewable Energy (IET-2), Erlangen, Germany
**3** Technische Universität Berlin, Institut für Theoretische Physik, Berlin, Germany

⋆ lc731@cam.ac.uk , † p.malgaretti@fz-juelich.de , ‡ holger.stark@tu-berlin.de

## Abstract

Isothermal information engines operate by extracting net work from a single heat bath through measurement and feedback control. In this work, we analyze a realistic active Szilard engine operating on a single self-propelled particle by means of steric interaction with an externally controlled mechanical element. In particular, we provide a comprehensive study of how finite measurement accuracy affects the engine's work and power output, as well as the cost of operation. Having established the existence of non-trivial optima for work and power output, we study the dependence of their loci on the measurement error parameters and identify conditions for their positivity under one-shot and cyclic engine operation. We also demonstrate that a suitably defined efficiency of information-to-work conversion, which at equilibrium is bounded above by unity as a consequence of Landauer's principle, may here be made arbitrarily large by increasing the active Péclet number of the particle. Equivalently, for such a nonequilibrium efficiency to remain bounded above by unity, the athermal motion of the bath particles needs to be accounted for explicitly. Notably, the information efficiency for one-shot operation exhibits a discontinuous transition and a non-monotonic dependence on the measurement precision. Finally, we show that cyclic operation improves information efficiency by harvesting residual mutual information between successive measurements.



# 1 Introduction

The possibility of extracting steady power from thermal fluctuations has been a topic of interest since Maxwell's seminal thought experiment, in which a "daemon" capable of sorting individual molecules according to their velocity would seemingly lead to a violation of the second law of thermodynamics [1]. The physical nature of information and its thermodynamic interpretation has thereafter undergone much scrutiny. Szilard's engines stand as particularly compelling illustrations of this intricate interplay. In one of the designs introduced in Szilard's original paper [2–4], a partition is inserted at the midplane of a box containing a single gas particle upon a binary measurement of the position of the particle relative to the midplane. To reduce the volume of the empty half of the box, a piston is slid in at no energetic cost, resulting, once the partition is removed, in an increase in free energy of the one-particle gas, which is subsequently converted into useful work via isothermal expansion. At thermodynamic equilibrium, this apparent breach of the second law of thermodynamics was reconciled by Landauer and Bennett's insights into memory writing and erasure [5,6]: upon expanding the phase space to include the daemon itself, the cyclic process of writing information, storing it, and eventually erasing it from physical memory gives rise to an irreversible auxiliary dynamics which is bound to generate entropy as a by-product [7–12].

An interesting extension of this framework has recently been explored in the form of Szilard engines coupled to active baths. Indeed, while the maximum work extracted from equilibrium thermal baths is limited by Landauer's principle, i.e. it is bounded from above by the thermodynamic cost of measurement, this limit is no longer valid for active baths [13–18]. Far from signalling a violation of the second law of thermodynamics, this work *surplus* is of course originated from diverting part of the underlying steady dissipation required to sustain the constitutive components of the nonequilibrium reservoir. Indeed, it is a characteristic feature of active engines that, when this additional source of dissipation is accounted for as an operational cost, quasistatic operation is outperformed by finite-time protocols [13,19].

While the energetic cost of measurement in information active engine can be quantified in much the same way as for the equilibrium case [17,20–24] and while its impact on the performance of information active engine has been under recent scrutiny [16,18], the nature of the coupling between the active substrate and the external system which extracts the work has so far been overlooked. In particular, the majority of existing works have focussed

on minimal models where forces are applied by exerting full control on the time-dependent form of an external potential (as generated, e.g., by an optical tweezer [16, 17, 25]) in which an active particle is confined. In contrast, work extraction in real-life microscopic active engines [26–28] is typically mediated by finite mechanical elements and relies on short-range steric interactions.[1] This detail, which at first glance may seem minor, leads to both quantitatively as well as qualitatively changes in the overall performance of the active engine in the presence of positional uncertainty. Intuitively, in a (nonlinear) external potential, positional error translates to an uncertainty in the force experienced by the active particle, and thus the instantaneous work rate; on the other hand, when manipulation is mediated by short-range forces, positional uncertainty leads to a variability in the encounter time, i.e. the waiting time before a constant force can be applied to the particle. Since work extraction in the second case can only occur for protocols of duration comparable to the persistence time, the interplay between these two timescales plays a nontrivial role, which we explore in the following.

Inspired by Szilard's design, we present here a minimal theoretical model capable of capturing these basic and crucial features. Accordingly, we revisit and expand upon the analysis of the "dynamic" active engine first introduced in Ref. [14], which extracts work by exploiting the finite correlation time (strong persistence) of the self-propulsion force exerted by a run-and-tumble (RnT) particle [16, 30]. At a constant rate, the demon measures the position and direction of motion of the particle; it then puts a piston ahead of the particle connected to a weight that opposes the motion of the particle. Now, if the piston is properly placed, after colliding with the piston, the RnT particle pushes against it and performs useful work by lifting the weight against the gravitational potential. However, due to errors in the measurement of position and velocity of the particle, the piston can be placed at a very unfavorable location so that the particle never meets the piston and negative work is done by the external force. This possibility, which was neglected for simplicity in Ref. [14], highlights the relevance of the measurement process in determining the performance of the active Szilard engine. In the following, we will demonstrate how to carefully analyze the former.

Our model shows that the work extracted attains an optimal value when the piston is initially located at a distance comparable to the measurement error in the position of the particle and for protocol durations comparable to the persistence time. Similarly, the information efficiency is also strongly affected by the precision of the positional measurement required to place the piston. This precision strongly determines both the measurement cost and the performance of the engine. Finally, we discuss the performance of the active Szilard engine in cycling conditions, i.e. under repeated use.

## 2 Model Setup

To keep the model simple, we assume the particle's motion to be restricted to one dimension. Accordingly, the active Brownian particle discussed in Ref. [14] is here replaced by a run-and-tumble (RnT) particle, most notably an idealized model of the motion of *E. coli* [31], whose self-propulsion velocity switches sign stochastically in the manner of a dichotomous noise [32]. In the presence of a generic external force $F_{\text{ext}}$ applied to the particle, the deterministic part of its velocity $v(t)$ is thus given by

$$v(t) = v_0 w(t) + \gamma_p^{-1} F_{\text{ext}}(t), \tag{1}$$

with $v_0$ the self-propulsion speed, $w \in \{-1, +1\}$ a symmetric dichotomous noise with (tumbling) rate $\alpha$ and $\gamma_p$ the friction coefficient. The persistence time and length for moving in

---

[1]Moreover, the power dissipated by auxiliary processes to establish a time-dependent virtual potential [25] may exceed the extractable power from the active substrate by many orders of magnitude [29].

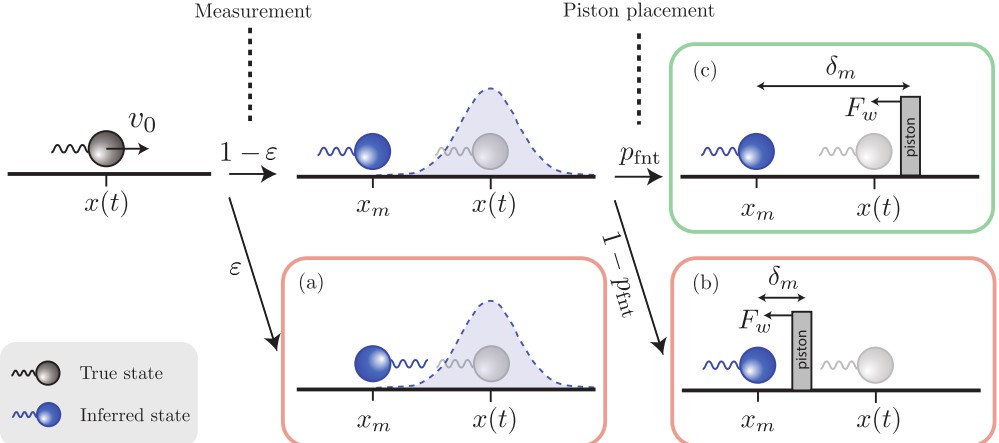

Figure 1: Work extraction protocol, shown schematically for a RnT particle with true internal state $w = +1$ at position $x$ at the time of measurement; (a), (b), and (c) show possible outcomes. Upon measurement, an estimate of the particle position $x_m$ and self-propulsion direction $w_m$ (indicated by the orientation of the "flagellum") is recorded, where $\epsilon$ is the probability to record the wrong direction. Based on the measurement outcome and given a particular value of $\delta_m$ a piston is placed at $x_m + w_m \delta_m$ and a force $F_w$ antialigned to the active velocity $v_0 w(t)$ is applied to it. The probability that $\delta_m$ is sufficiently large to correctly place the piston in front of the particle is $p_{\text{fnt}}$. The key model parameters are summarised in Appendix A.

one direction are thus $\tau_M = 1/\alpha$ and $d_M = v_0/\alpha$, respectively. Via the Stokes-Einstein relation, we have that $D_p = \gamma_p^{-1} k_B T$. We mostly work in the limit of large Péclet number, $\text{Pe} \equiv v_0 d_M / D_p \gg 1$, where particle translational diffusion ($D_p$) can be neglected on time scales comparable to $\tau_M$.[2] The trajectories are determined by the Langevin equation

$$\dot{x} = v + \zeta, \tag{2}$$

where $v(t)$ is given in Eq. (1), and $\zeta$ is a zero-mean Gaussian white noise with covariance $\langle \zeta(t) \zeta(t') \rangle = 2 D_p \delta(t - t')$.

The particle interacts with a piston which is initially placed at a distance $\delta_m$ from the measured position and then removed deterministically after a time $\tau$, as illustrated in Fig. 1. This setup is radically different from the cases where the work is extracted by means of external virtual potentials, such as optical traps. In fact while these potentials are present everywhere, the current setup is local and this introduces the possibility of "misses" (Fig. 1, red boxes), as well as the new length scale $\delta_m$. Upon placement, a force $F_w$ is externally applied to the piston, which, when not in contact with the particle, will then move at constant speed $v_w = F_w/\gamma_w$ in the direction of the force, where $\gamma_w$ denotes the friction coefficient of the piston. During periods of free motion of the piston, the external operator injects work in the system, which is immediately dissipated due to friction at a rate $\dot{W} = -F_w^2/\gamma_w$. On the other hand, when the particle and the piston are in contact with each other with the corresponding forces oriented in an antiparallel manner, they move together with net velocity $v_{wp} = (\gamma_p v_0 - F_w)/(\gamma_p + \gamma_w)$. In the case where $\gamma_p v_0 > F_w$, the active particle will then perform work against the external

---

[2]For reference, the typical run speed of *E.coli* is $v_0 \approx 15\ \mu\text{m} \cdot \text{s}^{-1}$ with tumbling rates of the order $\alpha \approx 1\ \text{s}^{-1}$ [33,34]. Approximating the cell structure as a sphere of radius $1\ \mu\text{m}$, whereby $D_p \approx 0.2\ \mu\text{m}^2 \cdot \text{s}^{-1}$ in water at room temperature, one finds $\text{Pe} \approx 10^3$. For diffusiophoretic Janus particles of similar size, $v_0 \approx 1\ \mu\text{m} \cdot \text{s}^{-1}$ while the persistence time $\tau_M \approx 10\ \text{s}$, whereby $\text{Pe} \approx 10^2 - 10^3$ [35].

force at an average rate

$$\dot{W} = \frac{F_w(\gamma_p v_0 - F_w)}{\gamma_p + \gamma_w}, \tag{3}$$

which is maximised for $F_w = \gamma_p v_0/2$ [14,16,36]. By considering the ideal scenario where the internal state is known and the optimal force is applied at all times, one then finds that the average extractable power is bounded above as $\dot{W} \le \dot{W}^* = \gamma_p v_0^2/[4(1 + \gamma_w/\gamma_p)]$. In the less ideal scenario where only the initial position is measured, we expect the extracted work to be maximized when the operating time of the piston, $\tau$, is comparable to the decorrelation time of the active agent, $\tau_M$. Indeed, for $\tau \ll \tau_M$, the piston is removed while positive work can still be extracted (at extremely short times the piston and the particle have not even come into contact); for $\tau \gg \tau_M$, there is a high probability that the RnT particle flips its velocity and leaves the piston.[3]

For simplicity, we take the post-measurement probability of the estimated particle position $x_m$ to be a Gaussian distribution centered at the true position $x$ and with standard deviation $\sigma_x$,

$$P(x_m|x) = \mathcal{N}(x_m; x, \sigma_x). \tag{4}$$

We note that $x_m$ is not the position at which the piston should be inserted. Indeed, as discussed below, placement at $x_m$ would result in a 50% probability of the piston being on the "wrong side" of the particle.

Clearly, the performance of such an engine is limited by the precision of the measurement protocol. Consider in particular, two possible occurrences where error-prone measurements can result in power loss, as illustrated schematically in Fig. 1: (a) the direction of self-propulsion is measured incorrectly, leading to the piston being placed "behind" the particle, which happens with probability $\varepsilon$; (b) the self-propulsion direction is measured correctly, but the measured position has a large error, leading once again to an incorrect placement. The likelihood of the second scenario, given by $(1 - \varepsilon)(1 - p_{\text{fnt}})$, can be reduced by placing the piston a distance $\delta_m$ ahead of the estimated position of the particle, cf. Fig. 1(c). However, this will also result in a longer period where piston and particle are not in contact, during which the power is purely dissipated into heat (or equivalently, negative power is extracted). Based on this argument, we expect that power extraction will be maximised for values of $\delta_m$ commensurate to the precision error $\sigma_x$.

## 2.1 Reduced units

In the following we rescale times by the persistence time $\tau_M = \alpha^{-1}$, lengths by the persistence length $d_M = v_0/\alpha$, forces by the stall force of the active particle $\gamma_p v_0$, and friction coefficients by $\gamma_p$. For ease of notation we keep the same symbols, so that we have

$$\begin{aligned} t/\tau_M &\to t, \quad x/d_M \to x, \\ F_w/\gamma_p v_0 &\to F_w, \quad \text{and} \quad \gamma_w/\gamma_p \to \gamma_w. \end{aligned} \tag{5}$$

Accordingly, the natural unit of work is $\gamma_p v_0 d_M$, i.e. the work done against viscous forces by an active particle self-propelling over one persistence length, and the natural unit of power is $\gamma_p v_0 d_M/\tau_M = \gamma_p v_0^2$, whence the reduced ideal power output as defined below Eq. (3) corresponds to

$$\dot{W}^* = \frac{1}{4(1 + \gamma_w)}. \tag{6}$$

---

[3]The tumbling event is particularly detrimental in the $1D$ model under consideration. For $2D/3D$ cases tumbling does not immediately imply losing contact with the piston.

Finally, the natural unit of efficiency, which we define below in Eq. (23), is given by the active Péclet number $\text{Pe} \equiv \gamma_p v_0 d_M / k_B T$ introduced earlier (equivalently, the reduced thermal energy scale is $\text{Pe}^{-1}$).

In the following two sections we explore the work output and the information efficiency of the active dynamic Szilard engine.

## 3  Average one-shot work output

Optimising the engine performance for a given measurement precision – as set by $\sigma_x$ and $\varepsilon$ – by fine tuning the control parameters $\tau$ and $\delta_m$ is a nontrivial problem. We address it here by first computing the average work $\overline{W}_{\text{os}}$ extracted under one-shot operation, i.e. the average work done on the piston after a single cycle of measurement, piston insertion and eventual removal. After accounting for the probability of the error scenarios described in the previous section and illustrated in Fig. 1, this is given by

$$\overline{W}_{\text{os}} = (1 - \varepsilon) p_{\text{fnt}} \overline{W}^+ + [\varepsilon + (1 - \varepsilon)(1 - p_{\text{fnt}})] \overline{W}^- . \tag{7}$$

Here, $\overline{W}^+$ and $\overline{W}^- \equiv -\tau F_w^2 / \gamma_w$ denote the average work extracted upon correct (green box in Fig. 1) and incorrect (red boxes in Fig. 1) placement of the piston, respectively. The contribution $\overline{W}^+$ will be calculated below.

### 3.1  Probabilities for determining $\overline{W}_{\text{os}}$

The probability $p_{\text{fnt}}$ of correctly placing the piston in front of the RnT particle depends on $\sigma_x$ and $\delta_m$ and is given, using Eq. (4), by

$$p_{\text{fnt}} \equiv P(x \leq x_m + \delta_m) = \frac{1}{2}\left[1 + \text{erf}\left(\frac{\delta_m}{\sqrt{2}\sigma_x}\right)\right], \tag{8}$$

where we assumed $w = +1$, i.e. rightward self-propulsion, without loss of generality.

The amount of work extracted in the case of correct placement depends on the true distance $\delta \equiv x_m + \delta_m - x$ between the particle and the piston immediately after placement, as well as on the time elapsed before the particle reverses its self-propulsion direction (i.e. tumbles). Thus, a further quantity needed in the calculation of $\overline{W}^+$ is the probability of the true distance $\delta$, conditioned on the piston having been placed on the correct side of the particle, i.e. on $\delta > 0$ in this case. This is given by

$$P_\delta(\delta | x \leq x_m + \delta_m) = \frac{2\mathcal{N}(\delta; \delta_m, \sigma_x)}{1 + \text{erf}\left(\frac{\delta_m}{\sqrt{2}\sigma_x}\right)}, \tag{9}$$

and is centered around the "buffer" distance $\delta_m$ specified by the protocol (for $\sigma_x = 0$, the piston is placed exactly at $x + \delta_m$).

The particle and the piston close the gap $\delta$ between each other in a typical time $\tau_\delta$, which we estimate from the deterministic part of the relative velocity between particle and piston, $F_w / \gamma_w + 1$ in reduced units, to be

$$\tau_\delta = \frac{\delta}{F_w / \gamma_w + 1} . \tag{10}$$

This time to encounter is distributed as

$$P_{\tau_\delta}(\tau_\delta) = \left(\frac{F_w}{\gamma_w} + 1\right) P_\delta\left(\left(\frac{F_w}{\gamma_w} + 1\right)\tau_\delta \middle| x \leq x_m + \delta_m\right), \tag{11}$$

where we used a straightforward transformation of probability from variable $\delta$ to $\tau_\delta$. We expect this distribution to be modified at small Pe due to the presence of translational noise $\zeta$ in Eq. (2). The latter will also contribute nontrivially to the effective pressure exerted by the active particle on the piston [14, 37, 38].

As already anticipated, increasing $\delta_m$ reduces the probability that the piston is placed incorrectly, however it also increases the time that the piston and the particle spend approaching each other. Note that, even upon correct piston placement, there is a finite probability that $\tau_\delta > \tau$, in which case particle and piston never come into contact during the work phase. This happens with probability $p_{\text{miss}}$ given by

$$p_{\text{miss}} \equiv \int_\tau^\infty d\tau_\delta P_{\tau_\delta}(\tau_\delta) = \frac{1 - \text{erf}\left(\frac{2\gamma_w\tau + \tau - 2\gamma_w\delta_m}{2\sqrt{2}\gamma_w\sigma_x}\right)}{\text{erf}\left(\frac{\delta_m}{\sqrt{2}\sigma_x}\right) + 1}. \tag{12}$$

We remark that the impossibility for piston and particle to instantly come into contact upon placement of the piston imposes an initial transient where work is lost to dissipation. This constitutes an inherent feature of our realistic design stemming from the finite range of steric interactions and the finite precision of the positional measurement.

Finally, we need to account for the fact that the particle may tumble during the work interval $\tau$. As me mentioned above, for $\tau \gg \tau_M$ the particle will undergo multiple tumble events and hence the power extracted will be very small or even negative (the piston does more work on the system than vice versa). Accordingly, we specialize to the case $\tau \lesssim \tau_M$, in which we can approximately neglect occurrences of more than one tumbling events during a period $\tau$. The probability density that a tumbling event occurs at a particular time $\tau_t$ (given in units of $\alpha^{-1}$) after the measurement of particle position and direction is given by the exponential law

$$P(\tau_t) = e^{-\tau_t}. \tag{13}$$

Now, for $\tau_\delta < \tau$, we have to consider separately the possibility that

(i) the particle tumbles before $\tau_\delta$, such that the particle and piston never meet. This happens with probability $p^{(i)} = 1 - e^{-\tau_\delta}$.

(ii) the particle tumbles at a time $\tau_t$ such that $\tau_\delta < \tau_t < \tau$, leading to a detachment from the piston while the protocol is still in progress. This happens with probability density $p^{(ii)} = P(\tau_t) = e^{-\tau_t}$.

(iii) the particle doesn't tumble during the duration of the protocol. This happens with probability $p^{(iii)} = e^{-\tau}$.

## 3.2 Calculating the mean work $\overline{W}_{\text{os}}$

We start with writing for a given value of $\tau_\delta$ the mean work extracted upon correct positioning of the piston as a combination of three terms corresponding to the cases (i)–(iii):

$$\overline{W}_{\tau_\delta < \tau}^+ = -\frac{\tau F_w^2}{\gamma_w}p^{(i)} + \int_{\tau_\delta}^\tau d\tau_t \left[-\frac{\tau_\delta F_w^2}{\gamma_w} + \frac{F_w(1-F_w)}{1+\gamma_w}(\tau_t - \tau_\delta) - \frac{(\tau - \tau_t)F_w^2}{\gamma_w}\right]P(\tau_t)$$
$$+ \left(-\frac{\tau_\delta F_w^2}{\gamma_w} + \frac{F_w(1-F_w)}{1+\gamma_w}(\tau - \tau_\delta)\right)p^{(iii)}. \tag{14}$$

In particular, the two negative terms under the integral describe the energy dissipated by the moving piston before it connects to the particle or after the particle has tumbled. The middle

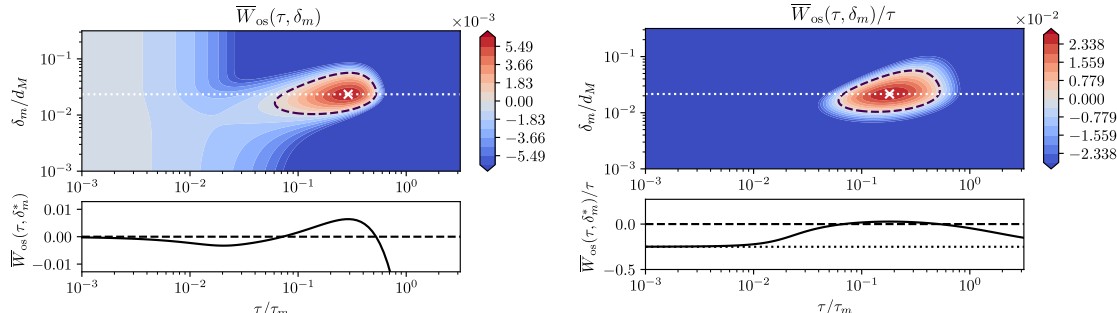

Figure 2: Average total work output (left) and work rate (right) of the dynamic Szilard engine under single cycle operation, plotted as a function of protocol duration $\tau$ and piston placement distance $\delta_m$. Plots under each contour show the work/power output as a function of $\tau$ along the line (white dotted) intersecting the respective optimum (white cross). Results are given in the units described in Sec. 2.1, fixing the error parameters $\sigma_x/d_M = 0.01$ and $\varepsilon = 0.1$, as well as setting $\gamma_w = 1$. The black dashed line indicates the zero-work/power level, such that inside the contour the average work/power is positive. The black dotted line corresponds to the power asymptote for vanishing $\tau$, Eq. (18).

term is the work performed by the particle on the piston. For $\tau_\delta > \tau$, particle and piston never get into contact and we have $\overline{W}^+_{\tau_\delta > \tau} = \overline{W}^-$. After simplifying Eq. (14) and setting $F_w = 1/2$ (in units of $\gamma_p v_0$) to maximise the power extracted during particle-piston contact (cf. Eq. (3)), one obtains:

$$
\begin{aligned}
\overline{W}^+_{\tau_\delta < \tau} &= \frac{1}{4\gamma_w}\left[\frac{(1+2\gamma_w)(e^{-\tau_\delta} - e^{-\tau})}{1+\gamma_w} - \tau\right] \\
&= \frac{\tau}{4\gamma_w}\left[\frac{(1+2\gamma_w)(1 - \frac{\tau_\delta}{\tau})}{1+\gamma_w} - 1\right] + \mathcal{O}(\tau^2, \tau_\delta^2),
\end{aligned}
\tag{15}
$$

where the last expression holds for $\tau_\delta < \tau \ll 1$. Using Eq. (15), $\overline{W}^+_{\tau_\delta}$ is thus positive when $\tau_\delta < \tau$ and

$$
\tau_\delta < -\ln\left[e^{-\tau} + \frac{\tau(1+\gamma_w)}{1+2\gamma_w}\right] = \frac{\tau\gamma_w}{1+2\gamma_w} + \mathcal{O}(\tau^2).
\tag{16}
$$

Since $\tau_\delta > 0$ by definition, the inequality can only be satisfied when $\tau$ is small enough such that the term in square brackets is smaller than unity.

The full average work $\overline{W}^+$ appearing in the right-hand side of Eq. (7) is then obtained by integrating Eq. (14) with respect to $\tau_\delta$ using the probability distribution of Eq. (11) and Eq. (12),

$$
\overline{W}^+ = p_{\text{miss}}\overline{W}^- + \int_0^\tau d\tau'_\delta\, \overline{W}^+_{\tau'_\delta} P_{\tau_\delta}(\tau'_\delta).
\tag{17}
$$

This integral can be performed to obtain an exact, albeit somewhat cumbersome, explicit expression for $\overline{W}^+$. Thus, after summing all relevant contributions according to Eq. (7), the average work output $\overline{W}_{\text{os}}$ is obtained. We report this latter expression in Appendix B.

### 3.3 Discussion of $\overline{W}_{\text{os}}$ and the mean work rate

The exact values of $\overline{W}_{\text{os}}$ as a function of the control parameters of the protocol, $\tau$ and $\delta_m$, while keeping error parameters $\sigma_x$, $\varepsilon$ fixed, are shown in Fig. 2. We observe two interesting features: first, the existence of a global maximum of the average work output. It occurs at

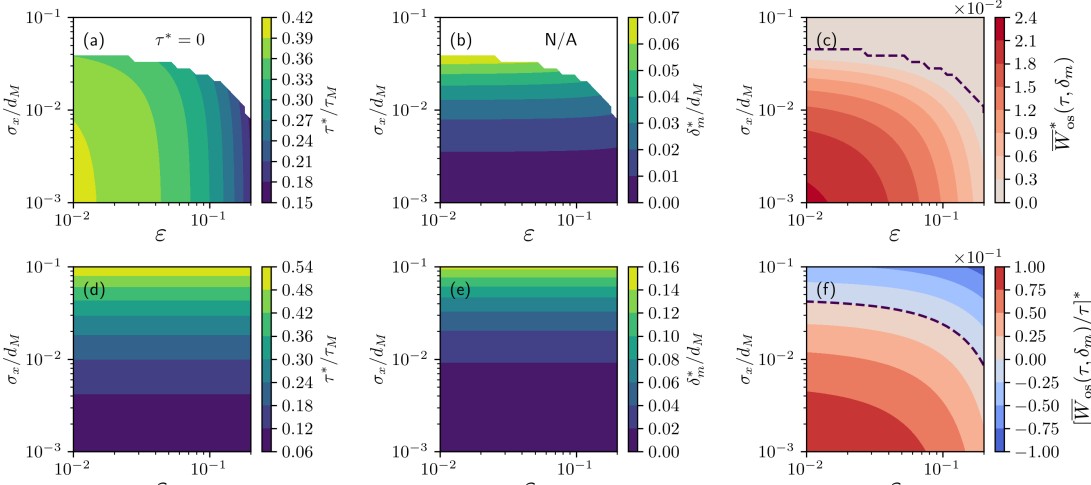

Figure 3: Dependence of the nontrivial maxima of work and power output on the error parameters, shown for $\gamma_w = 1$. (a,b) Dependence of the work optimal protocol control parameters $\tau^*$ and $\delta_m^*$ on $\varepsilon$ and $\sigma_x$. (c) Work output at the global maximum, shown as a function of the two error parameters. Since there exists always a line of protocols $\tau = 0$ for which $\overline{W} = 0$, we have a sudden transition to the trivial protocol being optimal at finite error (dashed line). (d,e) Dependence of the power optimal protocol control parameters $\tau^*$ and $\delta_m^*$ on $\varepsilon$ and $\sigma_x$. (f) Power output at the global maximum, shown as a function of the two error parameters. Recall that, in our rescaled units, the ideal scenario corresponds to $\dot{W}^* = 0.125$, which is obtained from Eq. (6) with $\gamma_w = 1$: as expected, this is the value the optimal power output converges to in the limit of vanishing error.

$\tau \simeq 0.3\tau_M$ and $\delta_m \simeq 0.02 d_M$ which, given the parameters used in Fig. 2, leads to $\delta_m \simeq 2\sigma_x$. This implies that the maximum work is obtained for protocols whose duration is somewhat smaller than the persistence time, $\tau_M$, and for pistons placed at a distance, $\delta_m$, which is larger than but comparable to the measurement error, $\sigma_x$. At the maximum, the work extracted is $\overline{W}_{os}/\tau\dot{W}^* \simeq 15\%$ of the ideal power output in the absence of errors, $\varepsilon = \sigma_x = 0$, and tumbling, as given in Eq. (6). Accordingly, the finite precision of the measurements strongly hinders the overall work extracted. The non-monotonic behavior observed in Fig. 2 is consistent with the arguments made above that performance should peak at intermediate values of the two control parameters. Finally, we remark that the extracted work is positive only in the close vicinity of the maximum whereas for $\tau = 0$ no work is done, $\overline{W}_{os} = 0$, hence leading to a local "trivial" maximum.

Another quantity of interest for information engines is the work rate, i.e. the power output, which is obtained by dividing $\overline{W}_{os}$ by the protocol duration $\tau$. Its dependence on $\tau$ and $\delta_m$ is akin to that of $\overline{W}_{os}$, except for the disappearance of the trivial maximum at $\tau = 0$. Indeed, in the limit of vanishing $\tau$, piston and particle never interact and we thus have the expected value

$$\lim_{\tau \to 0} \overline{W}_{os}(\tau, \delta_m)/\tau = -\frac{F_w^2}{\gamma_w} = \overline{W}^-/\tau, \tag{18}$$

which is strictly negative. Note that the power optimum generally does not co-locate with the work optimum.

Having established the existence of non-trivial optimal protocols for the work and power output, we proceed to study their dependence on the error parameters $\varepsilon$ and $\sigma_x$. As shown in Figs. 3(a) and 3(b), the location of the work optimum depends nontrivially on the measure-

ment precision. In particular, we observe that the optimal protocol duration, $\tau^*$, decreases upon increasing $\varepsilon$ and $\sigma_x$. This makes sense since larger measurement errors in location and direction reduce the possibility of persistent motion and thus the extraction of work. In contrast, the optimal choice of $\delta_m$ decreases slightly upon increasing $\varepsilon$ and increases noticeably with increasing $\sigma_x$, again compensating for the larger measurement errors. The average work output for the optimum protocol, Fig. 3(c), decreases upon increasing either of the errors, as expected, eventually crossing zero at finite values of $\varepsilon$ and $\sigma_x$. From this point onward, the nontrivial local maximum stops being the work optimum, as it becomes negative, and it is outperformed by the trivial protocol $\tau = 0$, where no work is extracted, $\overline{W}_{\mathrm{os}} = 0$, and for which $\delta_m^*$ is ill-defined.

As for the power optimum, Figs. 3(d)-(f) illustrate the nontrivial feature that the protocol parameters $\tau^*/\tau_M$ and $\delta_m/d_M$ are in this case independent of the directional error $\varepsilon$, while they both increase with $\sigma_x$. To understand this difference between work and power optima, we return to the definition of the one-shot power output $\overline{W}_{\mathrm{os}}/\tau$, which draws on Eq. (7). The optimum $(\tau^*, \delta_m^*)$ is defined by the extremisation condition

$$\partial_\tau [\overline{W}_{\mathrm{os}}/\tau]|_{\tau^*, \delta_m^*} = \partial_{\delta_m} [\overline{W}_{\mathrm{os}}/\tau]|_{\tau^*, \delta_m^*} = 0. \tag{19}$$

Noticing that $\overline{W}^-/\tau = -F_w^2/\gamma_w$ is independent of both $\tau$ and $\delta_m$, one can pull the common factor $1 - \varepsilon$ in Eq. (7) out of the extremisation condition, confirming that the power optimum does not depend on $\varepsilon$. Comparison of panels (a) and (d) also illustrates a striking difference between the dependence of $\tau^*$ on $\sigma_x$ for work vs power optima, which we ascribe to the additional $\tau^{-1}$ prefactor in the definition of the power comparatively penalising long cycles. Finally, the power at the optimum also decreases with both $\sigma_x$ and $\varepsilon$, as expected, and eventually it crosses the zero line in the high error regime, Fig. 3(f). Unlike the work, the power output at the optimum does not vanish beyond this point and instead becomes negative. This difference stems from the fact that the work output vanishes in the limit of zero protocol duration, $\lim_{\tau \to 0} \overline{W}_{\mathrm{os}} = 0$, while the power tends to a finite (negative) values in the same limit, $\lim_{\tau \to 0} \overline{W}_{\mathrm{os}}/\tau < 0$, which is moreover a lower bound to $\overline{W}_{\mathrm{os}}/\tau$ at finite $\tau$, cf. Eq. (18) and Fig. 2. As a result, in the high error regime, the engine operator may no longer "cut their losses" by setting $\tau = 0$ and are instead forced to inject more power via the piston than they can extract from the active particle, even at the optimum.

To summarize, the loci of both the work and power optima are consistent with the expectation that $\sigma_x/d_M < \delta_m^*/d_M \ll 1$ and $\tau^*/\tau_M \lesssim 1$, ensuring that on average piston and particle come and stay in contact for a substantial fraction of the persistence time, while maintaining the probability of incorrect piston placement sufficiency small. The relatively weak dependence of the loci of the optima on the error parameters observed in Fig. 3(a,b,d,e) indicates that optimal protocols are fairly robust to changes in the latter, i.e. no fine-tuning is needed.

# 4 Information efficiency

So far we dealt with the extracted work and associated power. However, a crucial observable for information engines is their "information efficiency", i.e. the ratio between the extracted work and the cost of measurement [16]. The thermodynamic cost of measurement arises because the act of gaining information reduces uncertainty about the system's state, as quantified by the change in mutual information $\Delta S$ between the system's true state and the measured state. This reduction in entropy directly corresponds to an energy scale via Landauer's principle, which states that the erasure or acquisition of information requires a minimum dissipation of $k_B T_m \Delta S$ as heat [17,20–23], with $T_m$ the working temperature of the measurement device. Naturally, higher measurement precision (smaller errors) will improve performance but also

result in a larger increase in mutual information, hence a higher operational cost for the information engine. In the following, we discuss the maximisation of the informational efficiency in light of this tradeoff, and more specifically the dependence of the efficiency on the measurement parameters $\varepsilon$ and $\sigma_x$, for two scenarios: one-shot and cyclic operation.

## 4.1 One-shot operation

We first consider the thermodynamic cost associated with a single cycle of measurement and piston insertion/removal. Assuming steady-state pre-measurement conditions, with uniform distribution of the active particle in an experimental domain of length $L$ and on both self-propulsion directions, $P(x, w) = 1/2L$, the simultaneous measurement of position and self-propulsion direction with error parameters $\sigma_x$ and $\varepsilon$ establishes a nonequilibrium state. The non-trivial correlations between the true state $(x, w)$ and the measured state $(x_m, w_m)$ is characterised by the conditional probability

$$p(x, w | x_m, w_m) = [(1 - \varepsilon)\delta_{w,w_m} + \varepsilon\delta_{w,-w_m}]\mathcal{N}(x_m; x, \sigma_x^2), \tag{20}$$

with $\delta_{a,b}$ the Kroeneker delta. The associated reduction in the differential Shannon entropy of $x, w$ given $x_m, w_m$, equivalently the mutual information between the true and measured states, is

$$\begin{aligned}\Delta S_{os}(\varepsilon, \sigma_x) &= S[P(x, w)] - S[p(x, w | x_m, w_m)] \\ &= -\ln\left(\frac{\sigma_x\sqrt{2\pi e}}{L}\right) + \ln 2 - s_\varepsilon.\end{aligned} \tag{21}$$

Here, we used $S[P(x, w)] = \ln 2 + \ln L$, defined

$$s_\varepsilon \equiv -\varepsilon\ln\varepsilon - (1 - \varepsilon)\ln(1 - \varepsilon), \quad 0 \le s_\varepsilon \le \ln 2, \tag{22}$$

and assumed $\sigma_x \ll L$ to approximate the entropy of the Gaussian distribution $\mathcal{N}(x_m; x, \sigma_x^2)$ by its form in unbounded space, i.e. $S[\mathcal{N}(x_m; x, \sigma_x^2)] \simeq -\ln(\sqrt{2\pi e}\sigma_x)$.

In rescaled units, the thermodynamic cost of measurement is then quantified formally as $Pe^{-1}\Delta S_{os}$, where we assume that the measured device is coupled to the same heat bath as the active particle. It diverges logarithmically as $\sigma_x/L \to 0$, e.g. when the precision error on the particle position goes to zero at fixed size of the experimental domain. The term $\ln 2 - s_\varepsilon$ in Eq. (21), on the other hand, is associated with measurement of the binary internal state and approaches the finite value $\ln 2$, familiar from the literature on the equilibrium Szilard engine, as $\varepsilon \to 0$.

We define the *information* efficiency

$$\eta_{os}(\tau, \delta_m, \varepsilon, \sigma_x) = \frac{\overline{W}_{os}}{Pe^{-1}\Delta S_{os}}, \tag{23}$$

as the ratio of the work extracted, in units of the thermal energy scale, over the information acquired by measuring the state of the particle. First of all, note that, since $\Delta S_{os}$ is independent of $\tau$ and $\delta_m$, the one-shot efficiency optimum at fixed measurement precision co-locates with the work optimum, $\eta_{os}^*(\varepsilon, \sigma_x) \sim \overline{W}_{os}^*$. Second, unlike $\overline{W}_{os}^*$, the optimal efficiency $\eta_{os}^*$ has a non-monotonic dependence on $\sigma_x$, see Fig. 4a, stemming from the abovementioned divergence of $\Delta S_{os}$ in the limit $\sigma_x \to 0$, while the extractable work is bounded above by Eq. (6).

The dependence of the information efficiency on the measurement precision can be better understood by plotting $\eta_{os}^*$ against the measurement information gain $\Delta S_{os}$, see Fig. 4b. For $\Delta S_{os}$ below a critical value (here around 5 nats, or 7 bits), $\eta_{os}^*$ is strictly zero. The efficiency then undergoes a continuous transition and grows upon further increasing $\Delta S_{os}$ since the work

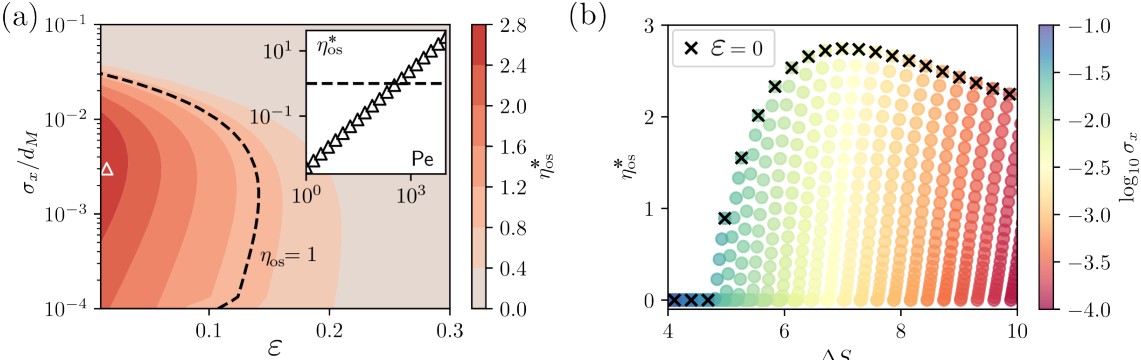

Figure 4: Cost of measurement and optimum efficiency under one-shot operation. (a) The efficiency shows a non-monotonic dependence of $\sigma_x$, while it is a monotonically decreasing function of $\varepsilon$. Inset: At fixed error parameters, exemplified by the white triangle in the main plot, the efficiency exceeds the Landauer bound for equilibrium information engines $\eta_{os} = 1$ (dashed line) for sufficiently large values of the Peclet number. (b) Plotting the same data for all $\epsilon, \sigma_x$ as a scatter plot of efficiency versus $\Delta S_{os}$ (the increase in mutual information upon measurement) shows a maximum efficiency at fixed $\Delta S_{os}$, which is achieved for $\varepsilon = 0$. A global maximum exists at finite $\Delta S_{os}$, indicating that beyond this point the additional cost of acquiring more detailed information exceeds the improvement in extracted work.

grows upon reducing the errors. Since $\eta_{os}^* \propto \overline{W}_{os}^*$, this transition has precisely the same origin as the one observed in Fig. 3(c) for the optimal work output. However, once sufficient information has been acquired – here, once the regime $\sigma_x/d_M \ll 1$ is achieved – further increasing the measurement precision is counterproductive as it increases the measurement cost without bounds with only marginal performance improvement, thus reducing the maximum accessible efficiency at high measurement information gain. The upper bound on $\eta_{os}^*$ as a function of $\Delta S_{os}$ corresponds to the directional-error-free limit $\epsilon = 0$ (crosses in Fig. 4b). Thus, while the existence of a continuous transition is inherent to our engine design, the non-monotonic dependence of $\eta_{os}^*$ on $\Delta S_{os}$ in the small error regime is a generic feature of active information engines relying on the measurement of a continuous variable, e.g. position.

We recall here that the Landauer bound for equilibrium information engines – which is nothing but a particular formulation of the second law of thermodynamics in that regime – may be written as $\eta_{os} \leq 1$ and is saturated by a Szilard engine operated quasistatically (see Appendix C for a Szilard-like setup with a Gaussian measurement posterior, as opposed to the single bit measurement of the "textbook" version). In the active regime, where highly persistent self-generated force fluctuations violate the standard fluctuation-dissipation relation [39], this bound no longer applies [14–18] and indeed we find that the information efficiency can be made arbitrarily large by increasing the Péclet number of the active particle (Fig. 4a, inset, where the line $\eta_{os}^* = 1$ is crossed at Pe $\simeq 10^3$). In particular, we find $\eta_{os}^* \sim$ Pe, as suggested by Eq. (23). We stress that this should not be interpreted as a violation of the second law of thermodynamics, which will of course apply to the total rate of entropy production, including the contribution originating from viscous friction between the active particle and the surrounding medium.

Finally, we remark that Eq. (23) is only one possible definition of efficiency in the active regime: since energy is continuously injected into and dissipated by the active particle at an average rate $\dot{\sigma}_{RnT}$ ($\dot{\sigma}_{RnT} = v_0^2/D_p$ for the free-particle case), one has the choice whether to consider this contribution to the total dissipation as an operational cost, in which case an alternative definition of the engine efficiency would be $\tilde{\eta}_{os} \equiv \overline{W}_{os}/(k_B T \Delta S_{os} + \tau \dot{\sigma}_{RnT})$. Here

we limit our discussion to $\eta_{\text{os}}$ as defined in Eq. (23) on the basis that (i) we want to draw a direct parallel with equilibrium information engines, where no such ambiguity arises and (ii) a rigorous treatment of the protocol-dependence of $\dot{\sigma}_{RnT}$ requires a thermodynamically consistent description of the self-propulsion mechanism [40–42], which goes beyond the simple RnT model considered in this work.

## 4.2 Cyclic operation

Typically, the information engine is operated cyclically at finite $\tau$. To address this more realistic operation scheme, we need to consider that the mutual information between the true state $(x, w)$ and the measured state $(x_m, w_m)$ need not have decayed to zero by the end of each cycle, its value itself being dependent on the cycle duration [17, 22, 24, 43]. Accordingly, the increase in mutual information upon refreshing of the measured state $(x_m, w_m)$, which we denote $\Delta S_\tau$, will be smaller than the one-shot value $\Delta S_{\text{os}}$ computed in Eq. (21), where $\Delta S_{\text{os}} = \lim_{\tau \to \infty} \Delta S_\tau$.

Denoting the measured state *pre-refreshing* as $(x_{m,0}, w_{m,0})$, the increase in mutual information upon refreshing is given by

$$\Delta S_\tau = S[P(x_\tau, w_\tau | x_{m,0}, w_{m,0})] - S[P(x_\tau, w_\tau | x_m, w_m)]. \tag{24}$$

The second term depends only on the assumed post-measurement distribution and is easily computed. Indeed, it is identical to the quantity appearing in the one-shot case, Eq. (21). The first terms requires more attention, since it is affected by the non-trivial dynamics of the RnT particle as it interacts with the piston. To estimate it, we make the following approximations:

(i) we ignore interactions with the domain boundaries, i.e. we assume $L/d_M \gg 1$ or, alternatively, a ring domain with periodic boundary conditions;

(ii) we assume that the typical time to encounter introduced in Sec. 3.1 is small relative to the protocol duration, $\overline{\tau}_\delta/\tau \ll 1$, and that the probability of piston misplacement is negligible, $p_{\text{fnt}} \lesssim 1$. Both conditions are satisfied near the optima identified in Figs. 3 and 4, where we found $\delta_m \simeq 2\sigma_x$ and $\tau \simeq 0.3\tau_M$, implying $p_{\text{fnt}} \simeq 0.97$, and $\tau_\delta/\tau \simeq 0.05$. As a result, for the purpose of characterising the probability density $P(x_\tau, w_\tau | x_{m,0}, w_{m,0})$, we may neglect the pre-encounter transient and subsume the interaction with the piston under a constant particle drift $-F_w/(\gamma_p + \gamma_w)$, which, conveniently, does not alter the Shannon entropy. The free particle friction coefficient is also effectively renormalised as $\gamma_p \to \gamma_p + \gamma_w$;

(iii) we neglect instances of more than one tumbling event during the period $\tau$, as we have done for the computation of the average work. This can be justified based on the expectation that relevant protocol durations are comparable to the persistence time $\tau_M$, as we indeed find below.

All together, these approximations allow us to neglect the protocol dependence of $\Delta S_\tau$, rendering the optimisation problem numerically tractable.

From the first approximation and choosing $x_{m,0} = 0$, $w_{m,0} = +1$ without loss of generality, we have, in the reference frame where the piston-induced drift vanishes,

$$P(x_\tau, w_\tau | x_{m,0} = 0, w_{m,0} = +1) = \mathcal{N}(x_\tau; 0, \sigma_x) * [(1 - \varepsilon)G_\rightarrow^\tau(x_\tau, w_\tau) + \varepsilon G_\leftarrow^\tau(x_\tau, w_\tau)], \tag{25}$$

where $G_\rightarrow^\tau$ and $G_\leftarrow^\tau$ denote the Green's function for a RnT particle initialised at $x = 0$ with internal state $w = +1$ and $w = -1$, respectively. Throughout this section, the asterisk ($*$) denotes a convolution, here involving the Gaussian distribution accounting for the imprecision

in the positional measurement. Based on the assumed symmetry of the RnT dynamics, the two Green's functions are related by

$$G_{\rightarrow}^{\tau}(x_{\tau}, w_{\tau}) = G_{\leftarrow}^{\tau}(-x_{\tau}, -w_{\tau}).$$

Since we specialize to the case of $t \ll \tau_M$, the Green's function for a RnT particle initialised in the right-moving state reads, in our rescaled units,

$$G_{\rightarrow}^{\tau}(x_{\tau}, w_{\tau}) = \frac{e^{\tau}}{1+\tau} \Bigg[ e^{-\tau} G_B^{t/(1+\gamma_w)}(x_{\tau} - \tau/(1+\gamma_w))\delta_{w_{\tau}, +1} + e^{-\tau} \int_0^{\tau} d\tau' \tag{26}$$

$$\times \int_0^L dx' G_B^{\tau'/(1+\gamma_w)}(x' - \tau'/(1+\gamma_w)) G_B^{(\tau-\tau')/(1+\gamma_w)}(x_{\tau} - x' + (\tau-\tau')/(1+\gamma_w))\delta_{w_{\tau}, -1} \Bigg]$$

$$= \frac{\mathcal{N}(x_{\tau}; 0, D_p \tau/(1+\gamma_w))}{1+\tau} * \left[ \delta(x_{\tau} - \tau/(1+\gamma_w))\delta_{w_{\tau}, +1} + \frac{\theta(\tau/(1+\gamma_w) - |x_{\tau}|)}{2}\delta_{w_{\tau}, -1} \right],$$

where the prefactor $(1 + \tau)^{-1}$ normalises properly for having neglected instances of more than one tumbling event. Here, $G_B^{\tau}$ denotes the Green's function of simple Brownian motion of duration $\tau$ with diffusivity $D_p/(1 + \gamma_w)$ for a particle initialised at $x = 0$, namely $G_B^{\tau}(x) = \mathcal{N}(x; 0, D_p \tau/(1+\gamma_w))$. The appearance of the factor $(1+\gamma_w)^{-1}$ reflects the renormalisation of the particle mobility as discussed in point (ii) above. Combining these, we eventually get

$$P(x_{\tau}, w_{\tau}|x_{m,0} = 0, w_{m,0} = +1) = \frac{\mathcal{N}(x_{\tau}; 0, D_p \tau/(1+\gamma_w) + \sigma_x^2)}{1+\tau}$$
$$* \Bigg[ (1-\varepsilon)\left( \delta(x_{\tau} - \tau/(1+\gamma_w))\delta_{w_{\tau}, +1} + \frac{\theta(\tau/(1+\gamma_w) - |x_{\tau}|)}{2}\delta_{w_{\tau}, -1} \right) \tag{27}$$
$$+ \varepsilon\left( \delta(x_{\tau} + \tau/(1+\gamma_w))\delta_{w_{\tau}, -1} + \frac{\theta(\tau/(1+\gamma_w) - |x_{\tau}|)}{2}\delta_{w_{\tau}, +1} \right) \Bigg],$$

which is correctly normalised with respect to double integration over $x_{\tau}$ and $w_{\tau}$. For short protocols, $\tau/\tau_M < 1$, the terms involving Heaviside theta functions in (27) can be neglected and, upon substituting into Eq. (24), we obtain (see Appendix D for a detailed derivation)

$$\Delta S_{\tau} = \frac{D_p \tau/(1+\gamma_w)}{2\sigma_x^2} + \tau(1-2\varepsilon)\ln\frac{1-\varepsilon}{\varepsilon} + \mathcal{O}(\tau^2). \tag{28}$$

The information gain thus shows a linear asymptote $\Delta S_{\tau} \sim \tau$ at small $\tau$, indicating that, while the cost per measurement vanishes in this limit, the cost "rate" becomes constant. Note that, similarly to the one shot case, the information gain per measurement diverges as $\sigma_x \to 0$ when all other parameters are kept constant.

The accuracy of the small $\tau$ approximation in Eq. (28) is illustrated in Fig. 5a by comparing with a numerical evaluation of Eq. (24) using the full expression for the conditional probability from Eq. (27). Note that, in general, $\Delta S_{\tau}$ will depend on the particle's level of activity, however this dependence drops out at leading order in $\tau$ since the short time scale dynamics are dominated by diffusion.[4]

---

[4]This is not in contradiction with the approximation made in Sec. 2, where quantities like the time $\tau_{\delta}$ needed for the particle to close the gap between itself and the piston were estimated based on the noise-averaged velocity. Indeed, for Pe sufficiently large, the dynamics may be effectively ballistic on timescales comparable to $\tau_M$, while being dominated by diffusion at very short timescales, as demonstrated by the scaling crossover in the mean-squared displacement observed in virtually all active particle models [44, 45].

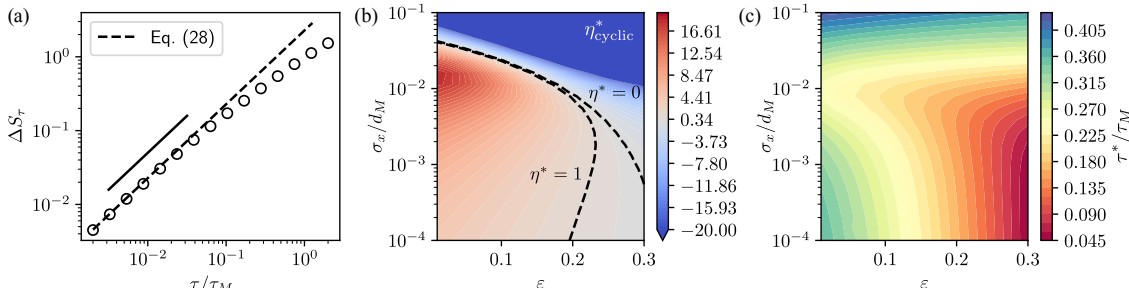

Figure 5: Efficiency at optimum for cyclic operation. (a) Comparison of exact (numerical) and approximate (analytical) results for the information gain. Solid line indicates linear asymptotic scaling. We set $\varepsilon = 0.1$, $\sigma_x = 10^{-2}$, $\gamma_w = \gamma_p = 1$ and $k_B T = 10^{-3}$. (b) Information efficiency at the optimum, calculated using approximate analytical expression, cf. Fig. 4a. Dashed curves indicate the Landauer-efficiency ($\eta = 1$) and zero-efficiency contours. (c) Protocol duration at efficiency optimum. The fact that $\tau^* < 1$ provides a self-consistency check for using the approximate expression in the evaluation of the cost.

Since the measurement cost $\mathrm{Pe}^{-1}\Delta S_\tau$ now depends explicitly on the protocol duration, the relation between efficiency and power/work optima is not trivial for cyclic operation. Indeed, in this case we cannot first optimise the work/power with respect to $\tau$ and $\delta_m$ but instead need to optimise the information efficiency for a given measurement precision,

$$\eta_{\mathrm{cyclic}}(\tau, \delta_m; \varepsilon, \sigma_x) \equiv \frac{\overline{W}_{\mathrm{os}}}{\mathrm{Pe}^{-1}\Delta S_\tau}, \tag{29}$$

as a whole. Using the approximate expression for the measurement cost in the small $\tau$ regime, Eq. (28), we thus compute the optimal efficiency $\eta^*_{\mathrm{cyclic}}(\varepsilon, \sigma_x)$ numerically. As shown in Fig. 5b, the dependency of the efficiency at the optimum on the error parameters $\sigma_x$ and $\varepsilon$ is qualitatively similar to that observed for the one-shot operation, see Fig. 4, even though the values tend to be higher in the case of cyclic operation, here by a factor 4-5. This is expected, as the cost of measurement is generally reduced compared to the one-shot case. We remark that, unlike the one-shot case (cf. Fig. 4), the efficiency at the optimum under cyclic operation becomes negative in the large error regime: this finding is analogous to the difference between the high error regimes of the work vs power output illustrated in Fig. 3(c,f) and stems from the denominator $\Delta S_\tau \sim \tau$ in Eq. (29) vanishing linearly in the trivial protocol limit $\tau \to 0$. In fact, in this regime, the efficiency optimum at fixed measurement precision co-locates with the power optimum. The self-consistency of the low $\tau$ approximation is verified by looking at the protocol duration $\tau^*$ at the efficiency optimum, Fig. (5)c, which is indeed found to be about one order of magnitude smaller than $\tau_M$.

While maintaining cyclic operation, one could in principle introduce a "refractory" timescale $\tau_{\mathrm{pause}}$ between the removal of the piston at the refreshing of the measurement. Indeed, for realistic applications in specific parameter regimes, the inter-measurement time $\tau^*$ determined by the optimisation procedure above may prove unrealistically small. While we don't treat this case explicitly here, we can nevertheless conclude that the magnitude of the efficiency $\eta_{\mathrm{cyclic}}$ can only decrease with $\tau_{\mathrm{pause}}$ since the numerator of Eq. (29) is independent of $\tau_{\mathrm{pause}}$ and the information gain is a monotonically increasing function of $\tau_{\mathrm{pause}}$, $\Delta S_{\tau + \tau_{\mathrm{pause}}} \geq \Delta S_\tau$ (cf. Fig. 5).

# 5 Discussion and conclusion

Building on previous work on active dynamic Szilard engines able to extract work from active baths [14,16], we considered a minimal design operating on a single active particle and explored the effects of finite measurement precision on its performance and efficiency. This design differs from other proposals, in that external control is applied via a finite mechanical element, rather than a time-dependent virtual potential [15,16,18,25]. This element of realism means that work cannot only be measured but also extracted. Since the mechanical coupling is a *local* mechanism (in contrast to an effective field defined everywhere), it introduces the additional length $\delta$, namely the distance at which the piston is placed in front of the particle. Our analysis shows that both $\delta$ and the parameters characterizing the active particle cooperate in determining the overall work extracted by the Szilard engine. In particular, our analysis shows the existence of nontrivial optima for both the work and power output. This underscores the ability of the engine to perform efficiently under non-equilibrium conditions (Fig. 2). However, these optima diminish significantly – eventually crossing zero – with increasing measurement errors (Fig. 3).

In order to assess the work extracted per unit information, we relied on the information efficiency, which provides a natural metric to quantify the conversion of measurement-derived information into useful work. While the performance of an equilibrium Szilard engine is limited by the Landauer bound ($\eta_{\text{os}} \leq 1$), we show that the information efficiency of our active dynamic engine can be made arbitrarily large by increasing the active Péclet number [14–18], see Fig. 4a. Interestingly, we find that a minimum information gain from the measurement is required to achieve a non-trivial optimum with finite efficiency (Fig. 4b).

Engines are typically used in a cyclic manner. Interestingly, our results show distinct differences between one-shot and cyclic engine operations. In the cyclic regime, residual mutual information between successive measurements led to a reduction in the effective cost of measurement, enabling comparatively higher efficiencies, see Fig. 5.

The observed robustness of the optima against moderate errors suggests that similar designs could be feasible in experimental systems employing real-time feedback control, such as those using video microscopy [46]. In that case, we might reasonably expect a finite delay $\tau_{\text{place}}$ between positional measurement and piston placement, which we haven't considered explicitly here. Nevertheless, for delays smaller than the diffusive-to-ballistic crossover timescale $D_p/v_0^2$ (approximately $10^{-3}$ s and $10^{-1}$ s for *E.coli* and Janus colloids, respectively [33,35]), such a delay effectively amounts to an increase in the error parameter $\sigma_x^2 \to \sigma_x^2 + D_p t_{\text{place}}$ and its impact on the engine performance may thus be characterised within the framework presented here (note that the cost of measurement will still be controlled by the unrenormalised error parameter). Finally, incorporating a more detailed description of the self-propulsion mechanism may provide deeper insights into the efficiency limits of active engines.

# Acknowledgments

**Funding information** LC acknowledges support from the Alexander von Humboldt Foundation.

# A  Model parameters

Table 1: List of model parameters appearing in this work.

|  | Symbol | Description |
|---|---|---|
| Active particle | $v_0$ | self-propulsion speed |
|  | $\alpha$ | tumbling rate |
|  | $\gamma_p$ | particle friction coefficient |
|  | $\tau_M = \alpha^{-1}$ | persistence time |
|  | $d_M = v_0 \tau_M$ | persistence length |
|  | $D_p = \gamma_p^{-1} k_B T$ | particle translational diffusivity |
| Wall | $F_w$ | external force on wall |
|  | $\gamma_w$ | wall friction coefficient |
| Protocol | $\sigma_x$ | positional error |
|  | $\varepsilon$ | orientatational error |
|  | $\delta_m$ | buffer distance for wall placement |
|  | $\tau$ | protocol duration |
|  | $\dot{W}^* = \frac{\gamma_p v_0^2}{4(1+\gamma_w/\gamma_p)}$ | maximum extractable work |
|  | $\delta$ | true particle-wall distance at placement |
|  | $\tau_\delta$ | waiting time for particle-wall encounter |

# B  Explicit evaluation of Eq. (17)

For the sake of completeness, we report below the explicit evaluation of the integral appearing in Eq. (17). This is given, in rescaled units, by

$$\overline{W}_{\mathrm{os}} = p_{\mathrm{miss}} \overline{W}^- - \frac{\tau\left(1 + \varepsilon + (\varepsilon-1)\mathcal{E}_1\right)}{8\gamma_w} - \frac{(\varepsilon-1)\left[C_1(\mathcal{E}_2 + \mathcal{E}_3) + C_2(\mathcal{E}_1 + \mathcal{E}_4 - 2)\right]}{8\gamma_w(1+\gamma_w)}, \quad \text{(B.1)}$$

where

$$
\begin{aligned}
C_1 &= (1 + 2\gamma_2)\exp\left(\frac{-2\gamma_w(\delta_m + 2\gamma_w\delta_m - \gamma_w\sigma_x^2)}{(1+2\gamma_2)^2}\right), \\
C_2 &= (1 + 2\gamma_w + \tau e^{\tau}(1+\gamma_w))e^{-\tau}, \\
\mathcal{E}_1 &= \mathrm{erf}\left(\frac{\delta_m}{\sqrt{2}\sigma_x}\right), \\
\mathcal{E}_2 &= \mathrm{erf}\left(\frac{2\gamma_w\delta_m - 2\gamma_w\sigma_x^2 + \delta_m}{\sqrt{2}(2\gamma_w\sigma_x + \sigma_x)}\right), \\
\mathcal{E}_3 &= \mathrm{erf}\left(\frac{2\gamma_w\left(2\gamma_w\left(-\delta_m + \sigma_x^2 + \tau\right) - \delta_m + 2\tau\right) + \tau}{2\sqrt{2}\gamma_w(2\gamma_w+1)\sigma_x}\right), \\
\mathcal{E}_4 &= \mathrm{erf}\left(\frac{-2\gamma_w\delta_m + 2\gamma_w\tau + \tau}{2\sqrt{2}\gamma_w\sigma_x}\right).
\end{aligned}
\quad \text{(B.2)}
$$

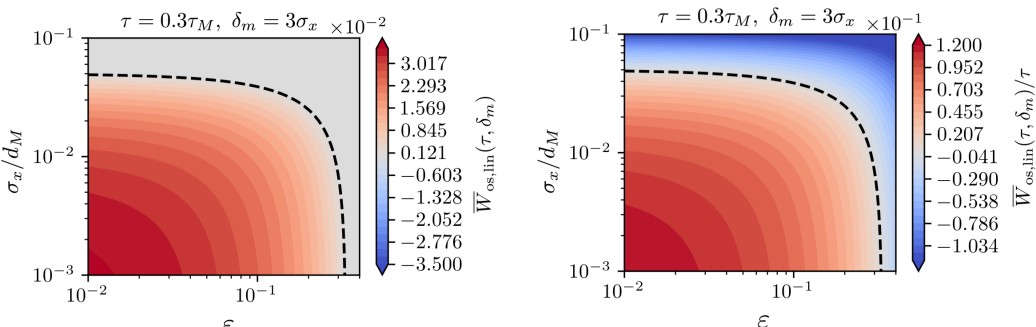

Figure 6: Contours for work and power as a function of the error rates obtained from the approximate expression Eq. (B.3) valid in the limit of small protocol duration (relative to the persistence time). We set $\tau/\tau_M = 0.3$ and $\delta_m/\sigma_x = 3$, based on the heuristics established in Sec.3 and the numerical findings shown in Fig. 3.

To recover a more interpretable solution we can replace $\overline{W}^+$ in Eq. (17) with its approximate form for $\tau_\delta < \tau \ll 1$, as given by the second equality in Eq. (15), whence we eventually obtain

$$\overline{W}_{\text{os,lin}} \simeq \frac{2\sigma_x \tau \left[(1-\varepsilon)\text{erf}\left(\frac{\delta_m}{\sqrt{2}\sigma_x}\right) - (1+\varepsilon)\right]}{16\gamma_w \sigma_x} - \frac{\sigma_x(1-\varepsilon)\left(e^{-\frac{\delta_m^2}{2\sigma_x^2}} - e^{-\Omega^2}\right)}{\sqrt{8\pi}(\gamma_w + 1)}$$
$$+ \frac{(1-\varepsilon)(2\delta_m - \tau)\left(\text{erf}(\Omega) + \text{erf}\left(\frac{\delta_m}{\sqrt{2}\sigma_x}\right)\right)}{8(\gamma_w + 1)} - \frac{1 - \text{erf}\left(2\Omega - \frac{\tau}{2\sqrt{2}\gamma_w \sigma_x}\right)}{4\gamma_w\left[\text{erf}\left(\frac{\delta_m}{\sqrt{2}\sigma_x}\right) + 1\right]}$$
$$+ p_{\text{miss}}\overline{W}^-, \tag{B.3}$$

where we defined the shorthand $\Omega \equiv [\gamma_w(\tau - \delta_m) + \tau]/(\sqrt{2}\gamma_w \sigma_x)$. The work and power output predicted by Eq. (B.3) are plotted in Fig. 6 as a function of the measurement error parameters for constant $\tau/\tau_M$ and $\delta_m/\sigma_x$. Comparing these results with the results of the protocol optimisation shown in Fig. 3 shows that the linearised theory provides an accurate picture of the error dependence in this system.

## C Landauer efficiency with a Gaussian posterior

We briefly illustrate how a Szilard-like equilibrium information engine operating quasistatically can achieve Landauer efficiency, $\eta_{\text{os}} = 1$ under one-shot operation, when the post-measurement distribution of the particle position is Gaussian, as we assume throughout this work. In particular, we consider the case where we are manipulating a Brownian particle with diffusivity $D_p$ confined in a 1d box of finite size $L$. Initially the position is unknown (uniform prior $P(x) = 1/L$), then a measurement is performed such that the post-measurement conditional is $P(x|x_m) = \mathcal{N}(x; x_m, \sigma_x^2)$ with $\sigma_x$ the precision error. (Here, we have assumed $\sigma_m \ll L$ so finite size corrections can be neglected.) Immediately after the measurement we apply the external harmonic potential $U(x; x_m) = \kappa(x - x_m)^2/2 - k_B T/2$ with $\kappa = 2D_p/\sigma_x^2$, chosen such that the posterior distribution corresponds to the Boltzmann measure associated with $U$, with average energy $\langle U \rangle = 0$. This is done with zero average work, since the average energy of the system does not change compared to the initial potential $U(x) = 0$. Now we

quasi-statically send $U(x) \to 0$, releasing the particle from the confinement and relaxing its distribution to the uniform prior. The associated work done against the potential is the difference in free energy $\Delta F$ between initial and final distribution. However, both states have the same average energy, so $W = \Delta F = \Delta S_{\mathrm{os}}$ reduces to the difference between the entropies of the initial distribution, $P_0 = P(x|x_m)$, and the final one (the uniform prior), as previously given in Eq. (21). The information efficiency is thus $\eta_{\mathrm{os}} = 1$ as in the usual Szilard engine, saturating the Landauer bound.

## D  Expansion of information gain

We start by rearranging the expression of the conditional probability, Eq. (27), as

$$
\begin{aligned}
P(x_\tau, w_\tau | 0, +1) = {} & \frac{\mathcal{N}(x_\tau; 0, D_p \tau/(1 + \gamma_w) + \sigma_x^2)}{1 + \tau} \\
& * \Bigg[ (1 - \varepsilon)\Big( \delta(x_\tau - \tau/(1 + \gamma_w))\delta_{w_\tau, +1} + \frac{\theta(\tau/(1 + \gamma_w) - |x_\tau|)}{2} \delta_{w_\tau, -1} \Big) \\
& \qquad + \varepsilon\Big( \delta(x_\tau + \tau/(1 + \gamma_w))\delta_{w_\tau, -1} + \frac{\theta(\tau/(1 + \gamma_w) - |x_\tau|)}{2} \delta_{w_\tau, +1} \Big) \Bigg] \\
= {} & \gamma_+ P_{+1}(x_\tau)\delta_{w_\tau, +1} + \gamma_- P_{-1}(x_\tau)\delta_{w_\tau, -1}, \qquad\qquad\qquad\qquad \text{(D.1)}
\end{aligned}
$$

where we grouped terms such that $P_{+1}(x_\tau)$ and $P_{-1}(x_\tau)$ can be read, respectively, as the normalised conditional probabilities of $x_\tau$ given $w_\tau = +1$ or $w_\tau = -1$. Accordingly, the normalisation factors $\gamma_+ \equiv P(w_\tau = +1)$ and $\gamma_- \equiv P(w_\tau = -1)$ are given by

$$
\begin{aligned}
\gamma_+ &= \frac{1 - \varepsilon}{1 + \tau}\left(1 + \frac{\varepsilon}{1 - \varepsilon}\tau\right) = (1 - \varepsilon)\left[1 + \frac{2\varepsilon - 1}{1 - \varepsilon}\tau + \mathcal{O}(\tau^2)\right], \\
\gamma_- &= \frac{\varepsilon}{1 + \tau}\left(1 + \frac{1 - \varepsilon}{\varepsilon}\tau\right) = \varepsilon\left[1 - \frac{2\varepsilon - 1}{\varepsilon}\tau + \mathcal{O}(\tau^2)\right].
\end{aligned}
\qquad\qquad \text{(D.2)}
$$

They satisfy $\gamma_+ + \gamma_- = 1$. Now recall the definition of the Shannon entropy

$$
S[P(x_\tau, w_\tau | x_{m,0} = 0, w_{m,0} = +1)] = -\sum_{\omega_\tau = \pm 1}\int dx\, P(x_\tau, w_\tau | 0, +1)\ln P(x_\tau, w_\tau | 0, +1), \quad \text{(D.3)}
$$

and the fact that for any $\gamma \in \mathbb{R}$

$$
S[\gamma P(x)] = -\gamma \ln \gamma + \gamma S[P(x)]. \qquad\qquad\qquad \text{(D.4)}
$$

Thus

$$
S[P(x_\tau, w_\tau | 0, +1)] = -\gamma_+ \ln \gamma_+ - \gamma_- \ln \gamma_- + \gamma_+ S[P_+(x_\tau)] + \gamma_- S[P_-(x_\tau)], \qquad \text{(D.5)}
$$

where, to linear order in small $\tau$,

$$
\begin{aligned}
\gamma_+ \ln \gamma_+ &= (1 - \varepsilon)\log(1 - \varepsilon) - \tau(1 - 2\varepsilon)[\log(1 - \varepsilon) + 1], \\
\gamma_- \ln \gamma_- &= \varepsilon \log \varepsilon + \tau(1 - 2\varepsilon)[\log \varepsilon + 1].
\end{aligned}
\qquad\qquad \text{(D.6)}
$$

Thus, using the notation introduced in Eq. (22),

$$
\gamma_+ \ln \gamma_+ + \gamma_- \ln \gamma_- \simeq -s_\varepsilon - \tau(1 - 2\varepsilon)\ln \frac{1 - \epsilon}{\epsilon}. \qquad\qquad \text{(D.7)}
$$

Let us now shift our attention back to Eq. (D.5). For moderate $\varepsilon$ and small $\tau$ such that $[\varepsilon/(1-\varepsilon)]\tau \ll 1$ and $[(1-\varepsilon)/\varepsilon]\tau \ll 1$, we have that $P_+ \simeq \delta(x_\tau - \tau/(1+\gamma_w))$ and $P_- \simeq \delta(x_\tau + \tau/(1+\gamma_w))$, whence

$$\gamma_+ S[P_+(x_\tau)] + \gamma_- S[P_-(x_\tau)] \simeq S[\mathcal{N}_{D_p\tau/(1+\gamma_w)+\sigma_x^2}], \tag{D.8}$$

is approximately the differential entropy of the convolved Gaussian distribution. Similarly, for $\varepsilon \to 0$ at fixed $\alpha\tau \ll 1$, we have $\gamma_+ \to 1$ and $\gamma_- \to 0$, whence we again obtain

$$\begin{aligned}\gamma_+ S[P_+(x_\tau)] + \gamma_- S[P_-(x_\tau)] &\simeq S[P_+] \simeq S[\mathcal{N}_{D_p\tau/(1+\gamma_w)+\sigma_x^2}] \\ &= \ln[2\pi e(D_p\tau/(1+\gamma_w) + \sigma_x^2)]/2\,.\end{aligned} \tag{D.9}$$

Overall, we obtain

$$S[P(x_\tau, w_\tau|0, +1)] \simeq s_\varepsilon + 2\tau(1-2\varepsilon)\ln\frac{1-\epsilon}{\epsilon} + \frac{1}{2}\ln(2\pi e\sigma_x^2) + \frac{1}{2}\ln\left(1 + \frac{D_p\tau/(1+\gamma_w)}{\sigma_x^2}\right). \tag{D.10}$$

Recalling that

$$S[P(x_\tau, w_\tau|x_m, w_m)] = \frac{1}{2}\ln(2\pi e\sigma_x^2) + s_\varepsilon\,, \tag{D.11}$$

and substituting Eqs. (D.10) and (D.11) into Eq. (24), we finally arrive at the expression for the increase in mutual information upon refreshing given in Eq. (28).

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
