# Peer review of "Mechanical work extraction from an error-prone active dynamic Szilard engine"

_SciPost Physics, doi:SciPost Phys. 19, 138 (2025)_

## Round 1 · Referee Report · Klaus Kroy (Referee 1) · 2025-6-26

Report

In their manuscript “Mechanical work extraction from an error-prone active dynamic Szilard engine” Luca Cocconi, Paolo Malgaretti, and Holger Stark aim to bring some useful element of reality into the discussion of active information engines. They report on what looks like a well-conceived and decently performed study with valid nontrivial (if unsurprising) results, and the manuscript is well written. Although I did not find any statement clarifying which of the “Expectations” criteria of the journal the authors aim to meet (sorry if I overlooked it!), I therefore feel that the manuscript could ultimately be published. I would however ask the authors to first address two issues concerning the motivation and physical interpretation.

First, I cannot imagine a large audience craving for realisations, let alone optimisations, of engines fired by information as useful practical devices for performing work. I therefore wonder whether it is really required to bring elements of reality to what is essentially, at its core, a pedagogical thought experiment best illustrated by some “minimal theoretical model”, as the authors call it, although theirs may not be the best (most minimalistic) choice.

Secondly, I am quite sceptical of the recent practice of motivating some otherwise less alluring afterthoughts about heat engines by the spectacular notion of a “violation” of thermodynamic bounds and, as a consequence, of the second law of thermodynamics. This has arguably caused quite some harm and confusion, calling for considerable efforts to clean up the mess. If a heat engine “violates” the second law (or whatever bound formalising it), this is the consequence of inconsistent thermodynamic definitions, typically involving inappropriately extended notions of “effective” (since they usually do not obey the zeroth law) temperatures and heat. A common mistake is to sell work-to-work as heat-to-work conversion. In particular, a random ensemble of microswimmers can serve as a heat bath only when one refrains from abusing individually targeted microswimmers as persistently driving motorboats in order to jump the line. Assigning a Peclet number to keep track of work-to-work conversion may help to quantify the fallacy but cannot cure it, of course. The only clean solution is to try and improve the generalised thermodynamic notions so that the second law, and meaningful bounds derived from it, are always obeyed.

In summary, to avoid reiterating elementary mistakes that have recently haunted the active-matter community, I would urge the authors to double check and enforce consistency with the fundamental notions of thermodynamics rather than “violating” them by presumably inappropriate definitions of heat and temperature (whenever they make sense, at all, these follow from the second law as explained in any thermodynamics class), and to ideally also think of a more convincing motivation for their work.

Recommendation

Ask for major revision

  • validity: -
  • significance: -
  • originality: -
  • clarity: -
  • formatting: -
  • grammar: -

Author:  Luca Cocconi  on 2025-09-03  [id 5774]

(in reply to Report 1 by Klaus Kroy on 2025-06-26)

We thank the referee for the effort and time they spent assessing our work. A diffTeX file highlighting changes to the manuscript is provided alongside this response.

Regarding their first point, we are not interested in optimisation per se, rather we are after the qualitative features of the optimal operational regimes as they emerge from that calculation. The model we chose is in fact one of the first models in the literature on active information engines exploiting the persistence of self-propelled particles, and part of what we are doing here is to carry out a more thorough analysis of it, highlighting details that were neglected in the original publication (Ref.[14]). For instance, the idea that placement of finite mechanical elements in space incurs by default a cost associated with the time to encounter and the associated tradeoffs were not considered there, while we show that this effect plays a non-negligible role. This is now discussed more extensively in the introduction:

This detail, which at first glance may seem minor, leads to both quantitatively as well as qualitatively changes in the overall performance of the active engine in the presence of positional uncertainty. Intuitively, in a (nonlinear) external potential, positional error translates to an uncertainty in the force experienced by the active particle, and thus the instantaneous work rate; on the other hand, when manipulation is mediated by short-range forces, positional uncertainty leads to a variability in the encounter time, i.e. the waiting time before a constant force can be applied to the particle. Since work extraction in the second case can only occur for protocols of duration comparable to the persistence time, the interplay between these two timescales plays a nontrivial role, which we explore in the following.

Regarding their second point, we note that a violation of Landauer’s equilibrium bound for out of equilibrium processes does not imply a violation of the second law of thermodynamics. We recognise that our original phrasing was somewhat misleading, in the sense that the bound does not apply to the regime of operation studied here and it is in that sense not violated. Our intention was to use the maximum Landauer efficiency as a meter to compare the efficiency of our engine and to make a point about the latter not being bounded from above. We are also not making any claims about effective temperatures or other such concepts, which we agree can be problematic. We have rephrased the discussion to clarify these points and further stress the fact that, of course, no violation of the second law of thermodynamics is implied by our results. See for example, in the updated abstract:

We also demonstrate that a suitably defined efficiency of information-to-work conversion, which at equilibrium is bounded above by unity as a consequence of Landauer's principle, may here be made arbitrarily large by increasing the active Péclet number of the particle.

Similarly, we now write in Sec.4.1:

We stress that this should not be interpreted as a violation of the second law of thermodynamics, which will of course apply to the total rate of entropy production, including the contribution originating from viscous friction between the active particle and the surrounding medium.

Attachment:

Scipost_diff_CR8aAiR.pdf

---

## Round 1 · Referee Report · Anonymous (Referee 2) · 2025-6-30

Report

The paper introduces a "realistic" model of an active Szilard engine that uses a single 1D run-and-tumble (RnT) particle as working medium. To differentiate their model from previous engine designs---mostly relying on externally imposed virtual potentials---here the particle interacts sterically with a piston, allowing measurement but possibly also extraction of mechanical work from the RnT particle. The authors explicitly incorporate measurement errors, both in position and orientation, and determine the protocol duration (close the particle's persistence time) and piston placement (comparable to measurement error) that optimizes the engine's work/power output under one-shot operation. To assess the work extracted per unit information the authors consider the information efficiency, which is found to exceed Landauer's bound in the limit of high Péclet; a common observation in active information engines. Lastly, they turn towards a cyclic operation of the engine and note an increase in information efficiency that is attributed to the retention of information between successive measurements. In this case, the optimal protocol duration is found to be one order of magnitude lower than the persistence time, to prevent the decay of mutual information between measurements.

While the paper is nicely written, easy to understand, and for the most part scientifically sound, I have some reservations that should be addressed by the authors before I can give a definitive judgment on whether or not the manuscripts should be accepted.

  • Throughout the manuscript the authors repeatedly emphasize their design to be realistic. While I agree that accounting for measurement uncertainties and a local mechanical coupling are necessary steps towards more realistic engine designs, they restrict themselves to a highly idealised case by studying a 1D RnT particle with binary orientation dynamics. Extrapolating the results to the physically meaningful higher dimensional case (or at least a more realistic 1D RnT particle with continuous reorientation angles) seems unfeasible. While technically sound, I thus have some reservation whether the present manuscript is of sufficient interest to motivate new research directions or to open new pathways. Indeed, the highlighted aspects of their model (particles interacting with a mechanical wall and the explicit account of measurement uncertainties) are not singular to the present work and can be found in existing literature on active information engines (including works from some of the authors).

  • Similar to earlier works, the authors report a violation of Landauer's bound for sufficiently high Péclet. From my understanding these violations are caused by an inadequate treatment of the systems thermodynamics when defining the efficiency. In particular, these studies systematically ignore the intrinsic dissipation that is required to maintain the bath particle's activity, leading to unphysically high efficiencies. While the authors briefly motivate the efficiency they used in the last paragraph of Sec.4.1, I frankly don't understand how this specific choice allows a meaningful comparison with equilibrium information engines. In my opinion, a violation of Landauer's bound should be reason enough to rule out an efficiency measure, especially when the authors strive for realism.

  • When turning towards the cyclic engine the authors neglect particle-boundary/piston interactions between measurements [point (i) on page 13]. According to the text, the authors are aware that this approximation only holds for a specific limiting case that is not guaranteed for the remainder of the section, yet apply it regardless. I strongly urge the authors to more carefully justify the use of this approximation (e.g., comparing it with numerical data, etc.,) instead of simply stating that "our results won't be exact". How large are the estimated deviations and is the reported behaviour robust?

Some more minor comments:

  • In the caption of Fig. 1 the authors write that the piston is placed at x_m + w_p \delta_m. I think it should be w_m instead.

  • There is a typo in lines 115-116: "...work can still be[ing] extracted..."

  • To improve clarity, I would recommend the authors to reconsider the symbols for the average extracted work for correct and incorrect placement of the piston. The overlap between superscripts (especially "-") and the bar makes it hard to discern them. It just looks like a longer bar.

  • In Fig.2 (bottom): What does the black dotted line represent?

  • There is a factor 1/\tau missing in the final equality of Eq.(18).

Recommendation

Ask for major revision

  • validity: -
  • significance: -
  • originality: -
  • clarity: -
  • formatting: -
  • grammar: -

Author:  Luca Cocconi  on 2025-09-03  [id 5775]

(in reply to Report 2 on 2025-06-30)

We thank the anonymous referee for the effort and time they spent assessing our work.

We address their three main reservations one by one as follows: 1)The design is indeed based on previous work and it is true that measurement uncertainty has been studied before in the context of active information engines. The novelty here is to resolve the complexities arising from the steric nature of the interaction (overlooked in the original paper) and explore their interplay with measurement error. Indeed, it is precisely when both are present together that the nontrivial features that we highlight come into play. This is now discussed more explicitly in the introduction:

This detail, which at first glance may seem minor, leads to both quantitatively as well as qualitatively changes in the overall performance of the active engine in the presence of positional uncertainty. Intuitively, in a (nonlinear) external potential, positional error translates to an uncertainty in the force experienced by the active particle, and thus the instantaneous work rate; on the other hand, when manipulation is mediated by short-range forces, positional uncertainty leads to a variability in the encounter time, i.e. the waiting time before a constant force can be applied to the particle. Since work extraction in the second case can only occur for protocols of duration comparable to the persistence time, the interplay between these two timescales plays a nontrivial role, which we explore in the following.

Similar to what was argued in Ref.[14], we can always imagine that the system is 2D/3D and we are just studying the dimension along the piston axis. We don’t expect that moving from RnT to a less analytically tractable dynamics for the self-propulsion force will modify our findings qualitatively.

2) We respectfully disagree. The information efficiency (both at and far from thermodynamic equilibrium) quantifies the efficiency of information-to-work conversion. It is only bounded above by unity when the second law of thermodynamics enforces this, however in general there is no reason why the work associated with a single bit measurement should be related in any way to the cost of measurement. It is true that an alternative efficiency measure could be constructed that quantifies the fraction of available work that is extracted through measurement. Whether the latter is more operationally meaningful depends on whether we consider the cost of “running the RnT particle” as an operational cost or not. In this case there is also an additional difficulty associated with computing the dissipation of the RnT particle under the action of the protocol. This is now discussed quite extensively in the manuscript, particularly towards the end of Sec.4.1:

In the active regime, where highly persistent self-generated force fluctuations violate the standard fluctuation-dissipation relation, this bound no longer applies and indeed we find that the information efficiency can be made arbitrarily large by increasing the Péclet number of the active particle (Fig.4a, inset, where the line $\eta_{\rm os}^* = 1$ is crossed at ${\rm Pe} \simeq 10^3$). In particular, we find $\eta^*_{\rm os}\sim {\rm Pe}$, as suggested by Eq.(23). We stress that this should not be interpreted as a violation of the second law of thermodynamics, which will of course apply to the total rate of entropy production, including the contribution originating from viscous friction between the active particle and the surrounding medium. Here we limit our discussion to $\eta_{\rm os}$ as defined in Eq. (23) on the basis that […] we want to draw a direct parallel with equilibrium information engines, where no such ambiguity arises.

3) Following the referee’s concerns about our argument for the simplifications introduced in Sec.4.2, we now justify these from a somewhat different angle with a broader range of validity (see list of approximations in Sec.4.2). In particular, rather than assuming free RnT motion between measurements, we instead consider that particle and wall remain in contact for most of the protocol duration, resulting in an effective drift and a renormalisation of the friction coefficient for the former. A more detailed description of the particle probability density dynamics under the action of the wall is highly nontrivial and beyond the scope of the current work, and would not qualitatively affect our main observation in this section, namely that cyclic operation can achieve higher information efficiency by leveraging residual mutual information.

As for the referee's minor comments: a) This was corrected. b) This was corrected. c) The notation was adjusted to avoid the overlap. d) This was clarified in the relevant caption. e) This was corrected.

---

## Round 1 · Referee Report · Anonymous (Referee 3) · 2025-7-20

Report

In this manuscript, the authors investigate a theoretically minimal model of an active dynamic Szilard engine operating on a single run-and-tumble particle. Considering a piston interacting sterically with the particle, they discuss a mechanism of work extraction via finite mechanical elements. Especially, the interplay between measurement errors (in both position and self-propulsion direction) and the thermodynamic performance of the engine is discussed. The authors identified non-trivial optima for both average extracted work and power output under one-shot operation. Calculating the information efficiency (work per unit information gain), they demonstrate that the Landauer bound is violated in the active regime. Finally, a comparison of engine performance is made between one-shot and cyclic operations, benefiting from residual mutual information.

This paper is technically solid and makes a clear, original contribution to the thermodynamics of information in active matter systems. It can be published in SciPost Physics with several clarifications and some extensions to the discussion. I suggest that the authors incorporate the following suggestions to improve accessibility and clarity of the manuscript.

(1) Please give some typical numbers of the important parameters such as tau_M, d_M, D_p for relevant biological systems or artificial systems. It would be useful to discuss whether the derived optimal parameters for work and power output (like tau*~0.3tau_M) are realistic in actual systems.

(2) As far as I understand, the model assumes the piston is placed immediately after measurement with no time delay. Can the authors discuss the effects of actuation delay (relative to tau_M) on the engine’s performance? A brief discussion or argument of this point would strengthen the applicability of the current model.

(3) Some plots like Fig. 3(f) and Fig. 5(b) show regions where the power or efficiency becomes negative. Although this is briefly discussed, a clearer physical interpretation would be valuable: Does this correspond to the engine being driven backward by the bath?

(4) I understand that the authors focus on the small Peclet number limit. It would be beneficial for the readers to discuss qualitatively the case when this condition is not satisfied. Some discussion of the effects of stochastic noise zeta in Eq. (2) might be useful.

(5) This work introduces several length and time scales. I sometimes got confused about the different meanings and definitions. A table summarizing all key parameters and variables (with some typical numbers) would enhance the clarity of the manuscript.

Recommendation

Ask for major revision

  • validity: -
  • significance: -
  • originality: -
  • clarity: -
  • formatting: -
  • grammar: -

Author:  Luca Cocconi  on 2025-09-03  [id 5773]

(in reply to Report 3 on 2025-07-20)

We thank the anonymous referee for the effort and time they spent assessing our work. A diffTeX file highlighting the changes to the manuscript is provided alongside this response. We have incorporated their suggestions and addressed their concerns one by one, as follows:

1) This information is now provided in a footnote on page 4. 2) This is now discussed in the conclusion:

In that case, we might reasonably expect a finite delay $\tau_{\rm place}$ between positional measurement and piston placement, which we haven't considered explicitly here. Nevertheless, for delays smaller than the diffusive-to-ballistic crossover timescale $D_p/v_0^2$ (approximately $10^{-3}$ s and $10^{-1}$ s for {\it E.coli} and Janus colloids, respectively, such a delay effectively amounts to an increase in the error parameter $\sigma_x^2 \to \sigma_x^2 + D_p t_{\rm place}$ and its impact on the engine performance may thus be characterised within the framework presented here (note that the cost of measurement will still be controlled by the unrenormalised error parameter).

3) Useful work is extracted by applying a force opposite to the ‘inferred’ active force. In the absence of mechanical coupling between particle and piston, there is no resistance force and the external force does negative work. The discussion on page 10 has been extended to repeat this point and hopefully making it clearer to future readers:

This difference stems from the fact that the work output vanishes in the limit of zero protocol duration, $\lim_{\tau \to 0} \overline{W}_{\rm os}=0$, while the power tends to a finite (negative) values in the same limit, $\lim_{\tau \to 0} \overline{W}_{\rm os}/\tau < 0$, which is moreover a lower bound to $\overline{W}_{\rm os}/\tau$ at finite $\tau$, cf. Eq.(18) and Fig.2. As a result, in the high error regime, the engine operator may no longer "cut their losses" by setting $\tau=0$ and are instead forced to inject more power via the piston than they can extract from the active particle, even at the optimum.

4) We added a comment about this on page 6, where we discuss estimating the time to encounter between piston and particle using the deterministic part of the velocity. The noise will also render the pressure exerted by the particle on the piston stochastic, complicating the treatment further:

We expect this distribution to be modified at small Pe due to the presence of translational noise $\zeta$ in Eq.(2). The latter will also contribute nontrivially to the effective pressure exerted by the active particle on the piston.

5) A table of key parameters has been added as a new Appendix A.

Attachment:

Scipost_diff.pdf

---

## Round 2 · Referee Report · Anonymous (Referee 3) · 2025-9-27

Report

The authors have responded appropriately to all of the comments and suggestions, and I consider this manuscript suitable for publication in SciPost.

Recommendation

Publish (easily meets expectations and criteria for this Journal; among top 50%)

---

## Round 2 · Referee Report · Klaus Kroy (Referee 1) · 2025-10-10

Report

The authors have revised and improved their manuscript in order to take into account the referees' suggestions, which makes it more suitable for publication. That said, I still think that the choice of the authors to wonder about the somewhat odd consequences of their own (in my view unfortunate) definition/normalization of the information efficiency is deplorable. I feel that it goes somewhat against the the spirit of thermodynamics and seems prone to promoting future misunderstandings of the sort that we have seen, recently, in the field of active heat engines -- like proclamations of even better information efficiencies obtained with the help of time tables for train departures (at least outside of Germany), say. While I do NOT insist on any further changes, I want to mention that, in place of statements like "We also demonstrate that a suitably defined efficiency of information-to-work conversion, which at equilibrium is bounded above by unity as a consequence of Landauer’s principle, may here be made arbitrarily large by increasing the active Péclet number of the particle" in the abstract, I would have preferred something of the sort "We also demonstrate that an appropriate technical definition of the efficiency of information-to-work conversion, which is bounded by unity via Landauer’s principle, has to account for the athermal motion of the bath particles".

Recommendation

Publish (meets expectations and criteria for this Journal)

  • validity: -
  • significance: -
  • originality: -
  • clarity: -
  • formatting: -
  • grammar: -

Author:  Luca Cocconi  on 2025-10-27  [id 5954]

(in reply to Report 2 by Klaus Kroy on 2025-10-10)
Category:
remark

We have included an additional sentence in the abstract to address this concern. In particular, it is now mentioned explicitly that for a nonequilibrium information efficiency to be bounded above by unity one would need to account for nonthermal motion of the bath particles as an operational cost. Having said this, we find our definition of efficiency (work extracted over operational cost of measurement by an observed interacting with the active bath) physically meaningful and sufficiently motivated in the manuscript.

---

## Round 2 · Author Response

We thank the referees and the editor for the effort and time they spent assessing our work. Their comments have helped us improve further our manuscript and we are hopeful that it may now be suitable for publication in SciPost Physics. We note that the main concerns raised by the three referees were of a conceptual nature, specifically with regards to physical motivation and interpretation of our results in light of general thermodynamic principles. Alongside some adjustments to notation and mathematical derivations (particularly in Sec.4.2), we have therefore focussed in our revision on clarifying these important conceptual points. Detailed responses to the individual reports are provided as comments to the original submission.

---

## Round 2 · List of Changes

Main changes: - Expanded discussion on physical motivation in the introduction - Clarify link between Landauer bound and the second law of thermodynamics out of equilibrium - Added a table of key parameters (Appendix A) and provided reference values (footnote 2) motivating our focus on the high Péclet regime - Added discussion of positional noise for particle-piston encounter dynamics - Added discussion of delay in piston placement - Revised Sec.4.2 to give a more general argument in support of the approximations applied there

---

## Round 3 · List of Changes

Edited the abstract in consideration of recommendation by Referee #2.

---

## Editorial Decision

published